# Association between vitamin D supplementation or serum vitamin D level and susceptibility to SARS-CoV-2 infection or COVID-19 including clinical course, morbidity and mortality outcomes? A systematic review

Amy Grove [1], Osemeke Osokogu,[1] Lena Al-Khudairy,[1] Amin Mehrabian [1,2]
Mandana Zanganeh,[1] Anna Brown,[1] Rachel Court,[1] Sian Taylor-Phillips [1]
Olalekan A Uthman [3], Noel McCarthy,[1] Sudhesh Kumar,[4] Aileen Clarke[1]

For numbered affiliations see end of article.

**Correspondence to**
Dr Amy Grove;
A.L.Grove@warwick.ac.uk

## ABSTRACT

**Objective** To systemically review and critically appraise published studies of the association between vitamin D supplementation or serum vitamin D level and susceptibility to SARS-CoV-2 infection or COVID-19, including clinical course, morbidity and mortality outcomes.

**Design** Systematic review.

**Data sources** MEDLINE (OVID), Embase (OVID), Cochrane Central Register of Controlled Trials, MedRxiv and BioRxiv preprint databases. COVID-19 databases of the WHO, Cochrane, CEBM Oxford and Bern University up to 10 June 2020.

**Study selection** Studies that assessed vitamin D supplementation and/or low serum vitamin D in patients acutely ill with, or at risk of, severe betacoronavirus infection (SARS-CoV, MERS-CoV, SARS-CoV-2).

**Data extraction** Two authors independently extracted data using a predefined data extraction form and assessed risk of bias using the Downs and Black Quality Assessment Checklist.

**Results** Searches elicited 449 papers, 59 studies were eligible full-text assessment and 4 met the eligibility criteria of this review. The four studies were narratively synthesised and included (1) a cross-sectional study (n=107) suggesting an inverse association between serum vitamin D and SARS-CoV-2; (2) a retrospective cohort study (348 598 participants, 449 cases) in which univariable analysis showed that vitamin D protects against COVID-19; (3) an ecological country level study demonstrating a negative correlation between vitamin D and COVID-19 case numbers and mortality; and (4) a case–control survey (n=1486) showing cases with confirmed/probable COVID-19 reported lower vitamin D supplementation. All studies were at high/unclear risk of bias.

**Conclusion** There is no robust evidence of a negative association between vitamin D and COVID-19. No relevant randomised controlled trials were identified and there is no robust peer-reviewed published evidence of association

## Strengths and limitations of this study

► The strengths of this systematic review include that it is reported in accordance with the Preferred Reporting Items for Systematic Reviews and Meta-Analyses (PRISMA) checklist.
► The review was conducted by two independent reviewers to ensure robustness of this work.
► We searched multiple living systematic review databases to enable us to capture publications in a fast-moving field of research.
► The limitations of the study relate to the small amount of evidence available which was at risk of bias and which limits the inferences that can be drawn.
► The review was restricted to the English language; therefore, non-English language papers may have been missed.

between vitamin D levels and severity of symptoms or mortality due to COVID-19. Guideline producers should acknowledge that benefits of vitamin D supplementation in COVID-19 are as yet unproven despite increasing interest.

## INTRODUCTION

COVID-19, a novel viral infection caused by SARS-CoV-2, was declared a pandemic by the WHO on 11 March 2020.[1] Mild COVID-19 may manifest as high temperature, a continuous cough and a loss of or change in sense of smell or taste.[2 3] However, more severe and critical cases can result in inflammation of the lungs, low oxygen levels and acute respiratory distress syndrome.[4] Interest is mounting regarding the association of vitamin D supplementation or level with susceptibility to COVID-19 due to the recognised modulating

effects of vitamin D on the immune system and immune response.

Vitamin D can modulate the immune system through highly expressed receptors in most non-skeletal tissues.[5 6] Two of the most common analogues of vitamin D which are found in food and used as a dietary supplement are $D_2$ (ergocalciferol) and $D_3$ (cholecalciferol, also made by the skin when exposed to sunlight).[7] Both $D_2$ and $D_3$ can be hydroxylated by liver enzymes CYP2R1 and CYP27A1 to form calcidiol (25(OH)D). The active metabolite of vitamin D, calcitriol (1α,25(OH)2D), results from the action of CYP27B enzyme on calcidiol. CYP27B is found in several tissues including the kidney, skin, bones and immune system.[8 9] Tumour necrosis factor-α (TNF-α) and interferon-γ (IFN-γ) are examples of inflammatory cytokines that stimulate the CYP27B enzymes of the immune system.[10–20] Vitamin D can interact with both the innate and the cellular immune systems through these mechanisms.

Current Public Health England (PHE),[21] National Institutes of Health[22] and European Food Safety Authority[23] recommendations highlight the importance of vitamin D to population health. Vitamin D deficiency is defined as less than 25 nmol/L (10 ng/mL) measured in blood serum.[21] The UK guideline recommendations suggest that people take a supplement of 10 µg of vitamin D per day during the winter months or throughout the year if they do not spend time outdoors or if they cover the majority of their skin when outside.[21] Published editorials, journal commentaries[24–29] and news media reports[30–32] suggest that individuals with low blood serum concentrations of vitamin D might be at higher risk of infection with COVID-19, or on infection have worse outcomes than individuals with normal/high serum vitamin D.[33]

Several observational studies have reported associations between low serum vitamin D and chronic[34] and acute conditions such as susceptibility to acute respiratory tract infections (RTI).[35–37] Most recently, Martineau and colleagues conducted a systematic review and meta-analysis of individual participant data from randomised controlled trials (RCTs) to assess the overall effect of vitamin D supplementation on risk of acute RTI.[38] They reported vitamin D supplementation to be safe while protecting against acute RTI overall (adjusted OR 0.88, 95% CI 0.81 to 0.96; p for heterogeneity <0.001). Patients very deficient in vitamin D benefited the most (adjusted OR 0.75, 0.60 to 0.95; p for interaction=0.006).[38] Critiques of this review have suggested that the findings should be interpreted as hypothesis generating only, as the results are heterogeneous and not sufficiently applicable to the general population.[39] Recent rapid reviews of vitamin D for treatment or prevention in COVID-19 reported no evidence that vitamin D deficiency predisposes to COVID-19, or that vitamin D supplementation is effective in prevention or treatment of COVID-19.[40 41] However, data sources included in the rapid review were limited.[42] Given the remaining uncertainty, it is timely to systematically review and critically appraise all peer-reviewed published evidence to assess the association of vitamin D supplementation or level with susceptibility to SARS-CoV-2 infection or COVID-19 including clinical course, morbidity and mortality outcomes.

## METHODS

### Protocol registration

The methods were prespecified in a protocol that was registered with the PROSPERO International Prospective Register of Systematic Reviews (https://www.crd.york.ac.uk/prospero/display_record.php?ID=CRD42020182876). Research ethics committee approval was not required for this study.

We undertook a systematic review to answer the following question: Is vitamin D supplementation or level associated with susceptibility to severe betacoronavirus infection (SARS-CoV, Middle East respiratory syndrome (MERS-CoV), SARS-CoV-2) including clinical course, morbidity and mortality outcomes?

Our review was conceptualised and written in accordance with the PRISMA statement.[43]

### Data sources and search

The search strategy was developed by the information specialists in collaboration with the research team and clinical advisors. We searched MEDLINE (OVID interface), Embase (OVID interface), Cochrane Central Register of Controlled Trials, MedRxiv and BioRxiv preprint databases on 6–8 May 2020. We searched the global research on COVID-19 developed by the WHO,[44] CEBM Oxford[45] and the living systematic review developed by Bern University[46] on 10 May 2020. We updated the database searches on 10 June 2020 to capture articles which may have been published since the initial search was conducted.

We searched additional resources including relevant systematic reviews (in MEDLINE (OVID interface), Embase (OVID interface) and Cochrane Database of Systematic Reviews, 19 May 2020), relevant references and contacted experts for additional evidence. Our full search record is included in the online supplemental file 1.

### Study eligibility

We developed predefined study eligibility criteria aligned to the research question (box 1). We imposed a date restriction of January 2002 to capture all published articles since SARS-CoV was first discovered in Asia in February 2003.[47] We limited to English language only.

### Article selection

Following the article search, we systematically identified and removed any duplicate citations using EndNote V.X9 software. Using titles and abstracts, deduplicated citations were screened by two independent reviewers (OO, MZ, AM, AG) and checked by a third (AC). All articles deemed ineligible were excluded at this stage. We

## Box 1   Study eligibility criteria

P—Population
1. Patients acutely ill with betacoronavirus infection (SARS-CoV, MERS-CoV, SARS-CoV-2).
2. Or at risk of acute illness with betacoronavirus infection.

I—Intervention/exposure
1. Vitamin D supplementation.
2. Low serum vitamin D.

O—Outcomes
1. Betacoronavirus infection (to include serological evidence of infection or clinically confirmed symptomatic infection).
2. Severe betacoronavirus infection (to include patients admitted to hospital or admitted to intensive care); mortality due to betacoronavirus infection.
3. Mortality due to betacoronavirus infection.

C—Comparators
1. No vitamin D supplementation.
2. High or normal serum vitamin D.

S—Study design
Peer-reviewed publications of randomised controlled trials and non-randomised studies were eligible for inclusion; including, non-randomised controlled trials, interrupted time series analyses, controlled before-and-after studies, cohort studies, ecological studies, case reports and case series.

Subgroups
1. Ethnicity characteristics (white British, all other white, mixed, Asian, black, other).
2. Age characteristics (population by 5-year age groups).

identified and obtained all remaining articles for full-text screening, which was performed independently by at least two reviewers against the prespecified eligibility criteria (box 1). Where disagreements regarding the inclusion of articles arose, a third reviewer (AC) was consulted to reach a final decision.

### Data extraction
Two reviewers independently (LA-K, MZ, OO, AM) extracted data from eligible full-text papers using a prespecified data extraction form. The accuracy of all the data extraction was independently assessed by a third reviewer (AG). Where reported, we sought to extract data from each article relevant to the research question, including details of population, intervention/exposure, comparator, outcomes and any detail related to the two prespecified subgroups: ethnicity characteristics and age characteristics. Disagreements between reviewers were resolved by discussion and agreement or via consultation with a third reviewer (AC).

### Risk of bias
The included studies had observational study designs aimed at answering a specific question. Therefore, risk of bias of included full-text papers was assessed using the Downs and Black Quality Assessment Checklist.[48] Two reviewers (AM, MZ, OO) independently assessed the risk of bias of the included studies and the accuracy of the assessment was evaluated by a third reviewer (LA-K).

### Data analysis
We anticipated that identified studies would be too heterogeneous to facilitate pooling of study data and planned a narrative synthesis. Nevertheless, we intended to consider pooling outcomes data in a meta-analysis using a random-effects model if appropriate.

### Patient and public involvement
Due to the rapid timeframe of this systematic review, it was not possible for our research team to involve patients or the public in the design, conduct or reporting of our study.

## RESULTS
After searching databases, assessing the reference lists of 17 narrative reviews[27 28 33 49–62] and one additional article identified through consultation with clinical experts,[38] we identified 499 citations. Following removal of duplicates and screening of titles and abstracts, we retrieved 59 full-text papers, of which 4 met the full eligibility criteria (see figure 1). The electronic supplement includes a list of reasons for excluding studies at full-text review. Seven articles closely met the eligibility criteria but were excluded as they were not available as peer-reviewed publications at the time of our narrative synthesis, details of these seven studies[63–69] is provided in the online supplemental material.

The characteristics of the four included studies are presented in table 1. All four included studies were conducted in Europe and published in April or May 2020. One study was based on data from UK residents exclusively,[70] another included data on residents in 20 European countries, including the UK.[71] The studies were observational design and no relevant RCT were identified or included in the review. All four studies were at high or unclear risk of bias and scored poorly across several domains of the Downs and Black Quality Assessment Checklist,[48] including external validity, internal validity and power. A prominent issue among the included studies was that the authors did not perform adequate multivariable adjustment to correct for confounding.[72–74] Ecological bias was present in Ilie *et al*,[71] which may result from spatial and temporal scale differences between country level mean levels of vitamin D. However, several domains in each risk of bias assessment were not applicable or not reported and, therefore, could not be scored using the Downs and Black Quality Assessment Checklist.[48] Detailed risk of bias scores are provided in the online supplemental material.

### Serum vitamin D
D'Avolio *et al*[73] used a cross-sectional design with data on nasopharyngeal swab PCR analysis for SARS-CoV-2 and a 25(OH)D measurement taken from patients between 1 March and 14 April 2020. PCR positives (median age=74 years (IQR 65–81); male=70.4%) had significantly (p=0.004) lower serum 25(OH)D levels (median=11.1

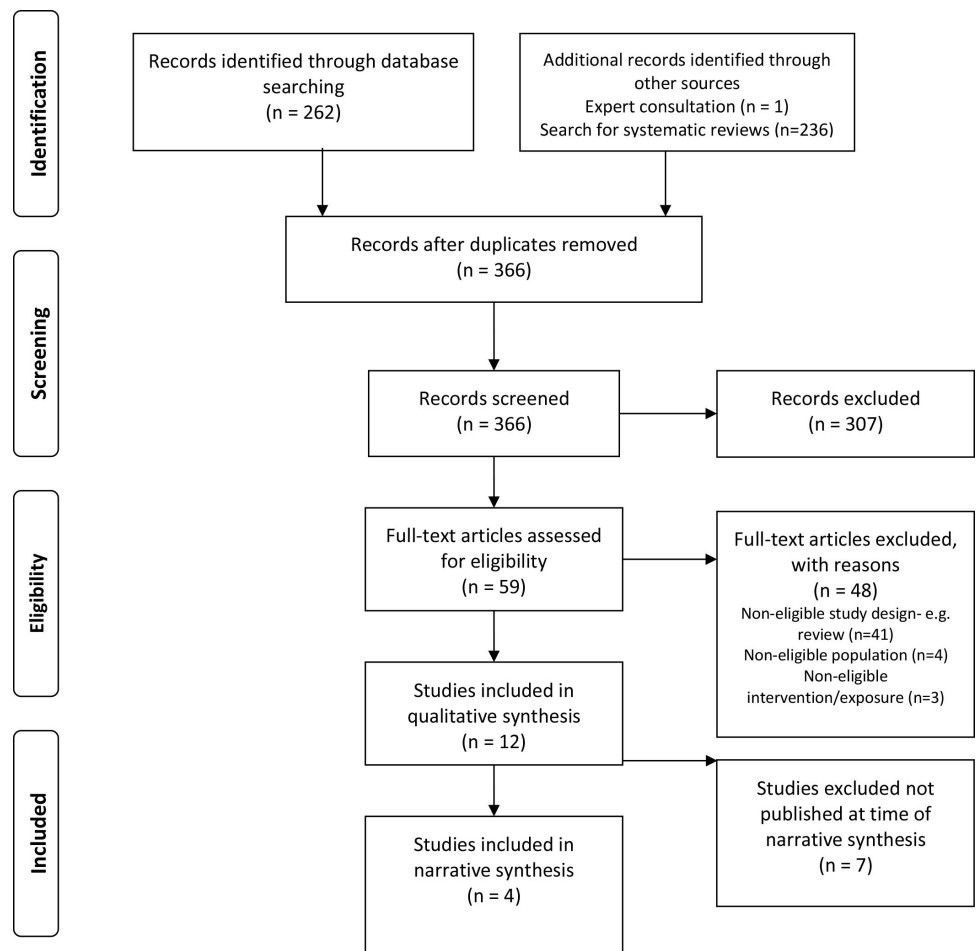

**Figure 1** PRISMA (Preferred Reporting Items for Systematic Reviews and Meta-Analyses) flow diagram for the selection of studies.

ng/mL (IQR 8.2–21.0)) than PCR negatives (median age=73 years (IQR 61–82); male=48.8%; median 25(OH)D=24.6 ng/mL (IQR 8.9–30.5)). Although gender-stratified and age-stratified analysis showed no significant differences, older (>70 years) SARS-CoV-2 positive (n=18) participants had significantly lower median serum 25(OH)D levels (9.3 ng/mL (IQR 8.1–19.9)) than older SARS-CoV-2 negatives (n=43) (23.1 ng/mL (IQR 8.5–31.7)) (p=0.037).

Hastie *et al*[70] is a retrospective cohort study that used data from the UK Biobank,[72] using data from 348 598 people with complete information on vitamin D and covariates; 449 people tested positive for COVID-19. COVID-19 positives were older (median=49 years; IQR=40–58) than COVID-19 negatives (median=49 years; IQR=38–57) with p value of <0.05. Multivariable analysis showed that age at assessment (OR=1.02; 95% CI=1.00 to 1.03; p=0.016) and non-white ethnicity (black OR=4.30, 95% CI=2.92 to 6.31, p=<0.001; South Asian OR=2.42, 95% CI=1.50 to 3.93, p=<0.001) were associated with confirmed COVID-19. There was no significant interaction between ethnicity and vitamin D deficiency (OR=0.90; 95% CI=0.66 to 1.23; p=0.515). Median vitamin D concentration at recruitment was lower for people with subsequent confirmed COVID-19 (28.7 (IQR 10.0–43.8) nmol/L) than for other

participants (32.7 (IQR 10.0–47.2) nmol/L) (p<0.01). Although univariable analysis suggested an association between vitamin D and COVID-19 (OR=0.99; 95% CI 0.99 to 0.999; p=0.013), this association became insignificant (OR=1.00; 95% CI=0.998 to 1.01; p=0.208) after adjustment for covariates.[70]

Ilie *et al*[71] used an ecological study design reporting on 20 European countries as at 8 April 2020; the data pertain to mean levels of vitamin D, cases of COVID-19 per million population and deaths from COVID-19 per million population. The authors performed Pearson's correlation coefficient calculations and reported a negative correlation between mean levels of vitamin D (mean 56.79 nmol/L, SD 10.61) and numbers of cases of COVID-19 per million population in each country (mean cases 1393.4, SD 1129.984, $r(20) = -0.44$; p=0.05). Additionally, a negative correlation was reported between mean vitamin D levels and the number of deaths caused by COVID-19 per million population in each country (mean 80.42, SD 94.61, $r(20) = -0.4378$; p=0.05). Sweden had the highest mean level of vitamin D (73.5 nmol/L) compared with Spain which had a mean level of 42.5 nmol/L. The number of cases of COVID-19 per million population was 834 in Sweden and 3137 in Spain. Likewise, at the time of the study, there were 68 deaths from

**Table 1** Characteristics of the four included studies

| Study | Design/setting | Population | Exposure/intervention | Outcomes | Results | Limitations |
|---|---|---|---|---|---|---|
| *Serum vitamin D* | | | | | | |
| D'Avolio et al[73] | Cross-sectional study Canton of Tessin, Switzerland | 107 patients with data on SARS-CoV-2 and 25(OH) D measurement | Vitamin D analysis, conducted within 7 weeks of the SARS-CoV-2 PCR result. Control patients with 25(OH)D data during the same period. | SARS-CoV-2 infection | Group 1 comprised 27 patients with positive PCR test results for SARS-CoV-2, while group 2 comprised 80 patients with a negative PCR result for SARS-CoV-2. Significantly lower 25(OH)D levels (p=0.004) in SARS-CoV-2 patients even after stratifying patients according to age >70 years. | Few patients from a single hospital. No available clinical information about the severity of COVID-19 symptoms. No data on other potential confounding variable SARS-CoV-2 and the 25(OH) D status were performed on different days. |
| Hastie et al[70] | Retrospective cohort study UK Biobank Cohort including England, Scotland and Wales | 502 624 participants aged 37–73 years between 2006 and 2010 | Biochemical assay of 25(OH)D, a measure of vitamin D status. Vitamin D was imputed if it was below or above the limit of detection. | Confirmed COVID-19 (at least one positive test result) | Complete data on 348 598 UK Biobank participants 449 had confirmed COVID-19. Of these, 385 (85.8%) were white compared with 64 (14.2%) non-white (black, South Asian and others). Vitamin D was associated with COVID-19 univariably but not after adjustment for confounders. Ethnicity was associated with COVID-19. | UK Biobank is not representative of the general population. Baseline measurements, including 25(OH)D concentration and health status, were obtained a decade prior to conduct of the study. |
| Ilie et al[71] | Ecological study 20 European countries | Population of 20 included European countries | Mean levels of vitamin D in each country | Cases of COVID-19 per 1 million population in each country. Deaths from COVID-19 per 1 million population. | Negative correlations between mean levels of vitamin D and the number of COVID-19 cases per 1 million, and mortality per 1 million. | The number of cases per country is affected by the number of tests performed and by the different measures taken by each country to prevent the spread of infection. |
| *Vitamin D supplementation* | | | | | | |
| Fasano et al[74] | Case–control survey. A single tertiary centre in Lombardy, Italy | 1486 Parkinson's disease (PD) patients were included in the survey 1207 family members (controls) | Vitamin D | 'Confirmed' or 'probable' diagnosis of COVID-19 | 12.4% of PD patients with confirmed or probable COVID-19 had been taking vitamin D. 22.9% of PD patients without COVID-19 had been taking vitamin D. | Well-known limitation of a telephone survey. Community-dwelling PD patients. Some patients could not be reached possibly due to death from COVID-19. COVID-19 diagnosis could not be confirmed in many cases. Younger age of non-PD COVID-19 cases. |

COVID-19 per million population in Sweden and 314 in Spain.

## Vitamin D supplementation

Fasano et al[74] investigated patients in a case–control phone survey in Lombardy, Italy. COVID-19 diagnosis was confirmed using a nasopharyngeal swab or probable based on (1) the presence of persistent COVID-19-related symptoms (≥3 including fever or ≥5 without fever); or (2) ≥1 symptom in the presence of suggestive chest radiologic signs; and/or (3) living with a family member with a confirmed diagnosis of COVID-19. A total of 1486 participants were included in the survey (32 confirmed COVID-19, 73 probable COVID-19 and 1381 unaffected). Confirmed/probable COVID-19 cases (mean age=70.5 (SD=10.1); male=53%) self-reported a significantly lower intake of vitamin D supplementation (12.4%) compared with unaffected cases (22.9%; mean age=73.0 (SD=9.5), male=57%). The age-adjusted OR (OR 0.56 (95% CI=0.32 to 0.99), p=0.048) suggested a protective effect of vitamin D intake.

## Subgroup evaluation

We planned to perform subgroup analyses by age and ethnicity. According to Hastie et al,[70] multivariable analysis showed that age at assessment (OR=1.02; 95% CI=1.00 to 1.03; p=0.016) and non-white ethnicity (black OR=4.30, 95% CI=2.92 to 6.31, p<0.001; South Asians OR=2.42, 95% CI=1.50 to 3.93, p<0.001) were associated with confirmed COVID-19. However, Hastie et al found no significant interaction between ethnicity and vitamin D deficiency (OR=0.90; 95% CI=0.66 to 1.23; p=0.515).

## DISCUSSION

This systematic review of non-randomised studies has shown no robust evidence of an association between vitamin D and COVID-19. We identified four studies for inclusion in a narrative synthesis which were all at high or unclear risk of bias. A univariable analysis of data from the UK Biobank database revealed an association between vitamin D and COVID-19 (OR=0.99; 95% CI 0.99 to 0.999; p=0.013). However, this association became insignificant (OR=1.00; 95% CI=0.998 to 1.01; p=0.208) after adjustment for 13 other covariates, suggesting that the initial association was due to one or more confounding variables.[70] This view is further strengthened by the demonstration of highly significant associations between age and ethnicity characteristics as predictor variables and COVID-19 as the outcome variable. Overall, the UK Biobank study showed no effect; however, it should be noted that the UK Biobank data included only one measurement of vitamin D levels taken between 10 and 14 years prior to the outbreak of COVID-19. This is a significant study limitation.

Liu et al[75] concluded that patients over 60 years experienced more severe manifestations and had longer disease courses of COVID-19 compared with patients

below 60 years.[75] And other studies have shown that older (rather than younger) people are more likely to die from COVID-19.[76–79]

Non-white people are known to be more susceptible to COVID-19 and tend to develop worse outcomes,[80] a finding that our review has further substantiated.[70] Ethnicity is a multifaceted construct that includes genetic makeup, sociocultural identity and behavioural patterns.[81] It has been shown to be associated with differing susceptibility and treatment outcomes in a number of diseases.[82–84] Hastie et al[70] did not find any interaction between ethnicity and vitamin D deficiency and although Ilie et al[71] identified a relationship, the study is subject to ecological bias. Ilie et al[71] compared vitamin D levels and rates of COVID-19 across 20 European countries, and therefore many relevant factors were not accounted for in the analysis. Given the findings so far from our review, we consider that there is paucity of data on vitamin D levels and morbidity and mortality from COVID-19 and there is no evidence from RCTs on outcomes of vitamin D supplementation on severity of symptoms or mortality to date. However, a relationship between ethnicity, vitamin D (serum levels or supplementation) and susceptibility to or severity of COVID-19 cannot yet be ruled out.

Risk of bias assessments demonstrate that all studies were at high or unclear risk of bias. All studies were observational designs and therefore, subject to confounding. The persistent calls for high-dose vitamin D supplementation[85] arise from speculation about presumed mechanisms.[86 87] Our systematic review found no robust evidence that low levels of vitamin D are associated with an increased likelihood of COVID-19. More robust prognostic studies could be combined in a systematic review where a prognostic factor research question is phrased, and considerations of participation, attrition, prognostic factor measurement, confounding measurement and account, outcome measurement, and analysis and reporting are evaluated.

Our systematic review identified no relevant RCTs; nevertheless, we are aware of two ongoing RCTs investigating the effects of vitamin D on COVID-19, the ZnD3-CoVici study, France (NCT04351490)[88] and the CoVitTrial, France (NCT04344041).[89] Both trials have an estimated study completion date of July 2020. Inclusion of data from these studies in future systematic reviews and meta-analyses may enable us to potentially draw better stronger conclusions on this topic. Results from the ongoing international VITDALIZE study (NCT03188796) may also contribute to our understanding of the effect of high-dose vitamin $D_3$ on mortality.[90]

## Study limitations

We performed a full systematic review of the published evidence available, and simultaneous independent screening, data extraction and risk of bias assessments. However, our study is limited by the small amount of evidence available which was, moreover, at risk of bias. This limits the inferences that can be drawn. Seven

eligible studies were excluded because they are not available as peer-reviewed publications.[63–69] If published, these seven studies would be included in a future update of this review. A final limitation is that the review was restricted to English language only. Therefore, articles published in other languages may have been excluded.

## Implications for practice

Our review does not provide evidence for or against additional or high-dose vitamin D supplementation specifically in relation to COVID-19. Treatment as standard practice for people who are deficient is pre-existing practice across Europe[23] the USA[22] and in the UK.[21] Current guidelines from PHE suggest that the entire UK population should take vitamin D supplements to prevent vitamin D deficiency in winter or with inadequate sunlight exposure to sun in summer.[21] This review does not give evidence to drive a change in this current advice. Treatment recommendations for patients should be updated following the publication of results from ongoing and new well-designed adequately powered randomised controlled trials.

## CONCLUSION

This systematic review identified no robust evidence to enable us to assess an association between vitamin D supplementation or serum vitamin D level with susceptibility to COVID-19 including clinical course, morbidity and mortality outcomes. All studies were at high or unclear risk of bias. Both age and ethnicity were associated with vitamin D levels even after multivariable adjustment. Black and South Asian people had a much higher risk of confirmed COVID-19 compared with white people. However, there was no interaction between the association of ethnicity and vitamin D deficiency with COVID-19. There were no papers reporting association of vitamin D with severity of symptoms or mortality due to COVID-19.

**Author affiliations**
[1]Division of Health Sciences, Warwick Medical School, The University of Warwick, Coventry, UK
[2]Department of Pharmaceutical Nanotechnology, University of Medical Sciences, Mashhad, Iran (the Islamic Republic of)
[3]Warwick-Centre for Applied Health Research (WCAHRD), University of Warwick, Coventry, UK
[4]Warwick Medical School, The University of Warwick, Coventry, UK

**Acknowledgements** We thank Kath Charters, Welsh Ambulance Services NHS Trust, for comments on the study design and drafts of this manuscript and for valuable advice and helpful discussions.

**Contributors** SK, AG and AC conceived the study. AG, AC, NM, SK, ST-P and OAU designed the study. RC and AB developed the search strategies, performed all searches and database management, and created the bibliography. AG, AC, AM, OO, MZ screened titles and abstracts for inclusion. AG, OO, AM, MZ, LA-K and AC screened at full text and extracted and analysed data. OO, AM, MZ and LA-K performed risk of bias assessments. AC, SK and NM assisted in the interpretation from a clinical perspective. ST-P, LA-K and OU offered technical and methodological support. AG, OO and MZ wrote the first draft, all authors revised content. All authors approved the final manuscript. AG and AC are the guarantors. The corresponding author attests that all listed authors meet authorship criteria and that no others meeting the criteria have been omitted.

**Funding** Dr AG is supported by the National Institute for Health Research (NIHR) Advanced Fellowship Programme Reference (AF-300060). Dr ST-P is supported by an NIHR Career Development Fellowship (CDF-2016-09-018). Professor AC and Dr LA-K are supported by the NIHR Applied Research Centre West Midlands (ARC-WM). All other authors, with the exception of Professor Uthman, Professor SK and Professor NMC, are supported by the NIHR, Evidence Synthesis Programme (HTA 14.25.05). The views expressed are those of the authors and not necessarily those of the NIHR or the Department of Health and Social Care.

**Competing interests** None declared.

**Patient consent for publication** Not required.

**Provenance and peer review** Not commissioned; externally peer reviewed.

**Data availability statement** Data are available on reasonable request. All data relevant to the study are included in the article or uploaded as supplementary information. The study protocol is available systematic review protocol registration: CRD42020182876 available online via PROSPERO at https://www.crd.york.ac.uk/prospero/display_record.php?ID=CRD42020182876. All included studies are publicly available. Additional data are available on reasonable request by emailing the corresponding author.

**ORCID iDs**
Amy Grove http://orcid.org/0000-0002-8027-7274
Amin Mehrabian http://orcid.org/0000-0002-8879-7565
Sian Taylor-Phillips http://orcid.org/0000-0002-1841-4346
Olalekan A Uthman http://orcid.org/0000-0002-8567-3081

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
