## [Reviewer comments · BMJ Open]

ARTICLE DETAILS

TITLE (PROVISIONAL)	Association between vitamin D supplementation or serum vitamin D level and susceptibility to SARS-CoV-2 infection or COVID-19 including clinical course, morbidity and mortality outcomes? A systematic review.
AUTHORS	Grove, Amy; Osokogu, Osemeke; Al-Khudairy, Lena; Mehrabian, Amin; Zanganeh, Mandana; Brown, Anna; Court, Rachel; Taylor-Phillips, Sian; Uthman, Olalekan; McCarthy, Noel; Kumar, Sudhesh; Clarke, Aileen

VERSION 1 – REVIEW

REVIEWER	Mark Bolland University of Auckland, New Zealand I've authored articles/meta-analyses on vitamin D supplementation, and generally found little effect of vitamin D supplements hence suggested against routine supplementation. Otherwise no financial conflicts
REVIEW RETURNED	03-Sep-2020

GENERAL COMMENTS	General comments: This is a highly relevant topic as the authors point out. The methods and review itself seem to be good quality. I found the Results and Discussion long and repetitive and have made some suggestions. Major points: 1. P6 L45- the Martineau meta-analysis (ref 35) is used in most pieces promoting vitamin D supplementation for Covid, including this one. So, it warrants closer analysis. Some points to consider: the adjusted odds ratio of 0.88 is misleading as it overestimates the risk reduction because RTI events were common. The relative risk reduction wasn't 12%, but closer to 6%, with an absolute reduction of 2-3% which both seem pretty trivial. Currently, one subgroup analysis has been highlighted, which is quite common in the pieces promoting vitamin D. But the other subgroup analyses show the benefits were in children not adults, and those given daily or weekly but not bolus dosing of treatment. All of these subgroup analyses were largely dependent on a couple of strongly positive trials. One in particular- a cluster RCT in Mongolian children- is especially influential. But clearly generalising results from children with colds to adults with Covid, or from countries like Mongolia/Afghanistan to the UK is not appropriate.
---

	These issues should be discussed in a little more detail than is currently there. We have written about this in our editorial and responses to the Martineau paper/editorial and more recently in the Irish Medical Journal in response to ref 48. Neither of our papers need to be cited but they do cover the issue in a lot of detail which might help the authors. The authors should highlight the problem with the inappropriate generalisation of these results, which unfortunately, but predictably, has come to pass with the current covid pandemic. This issue becomes critical given the statement “If vitamin D supplementation could influence the susceptibility of people to COVID-19 infection, the intervention could be cost-effective with few associated safety risks.” in p7 L7 where the Martineau results are used as the basis for suggesting vitamin D supplementation might be beneficial for covid. In my view, the Martineau results don’t suggest any likely benefit once they examined at anything more than a superficial level. 2. Since policy about vitamin D supplements and covid will be best informed by RCTs, the Results should start with an explicit statement that no relevant RCTs were identified eg somewhere in the first paragraph, and this should also be mentioned explicitly in the Discussion and Abstract 3. Much of the narrative detail in the Results can be presented much more simply in a Table similar to the table of study characteristics in the supplement. This would allow the key points related to vitamin D and covid only to be presented in the text, and the Results section could then be much briefer and more focussed. 4. The data from the studies is incredibly weak as the authors discuss, but I think that needs to be highlighted more. The UK biobank study showed no effect. The Swiss study didn’t present sufficient data to know whether the groups were comparable. The Italian case-control simply records vitamin D supplement usage, not even 25OHD levels. The ecological study is largely worthless as it only compares vitamin D levels and covid rates at a country level when obviously far more important factors are not considered. Rather than repeating the results in the first paragraph of the Discussion, I suggest the authors synthesize them. My interpretation of the Results is that there is no robust evidence of any link between 25OHD and covid. 5. Paragraphs 2-4 of the Discussion are largely redundant because the findings are either neutral or not robust. This section can be shortened substantially or deleted. 6. P13, L33 – there is no evidence, not “limited” evidence isn’t there? Likewise in the Abstract conclusions: the first sentence differs from the second and seems both unnecessary and inconsistent with the actual findings of no reliable evidence rather than “limited” evidence. 7. P13, L44-52 – I don’t understand the relevance of this text, or why speculation about vitamin D’s effects is appropriate in the absence of evidence of effect.
--	---

	8. Implications for practice section- presenting the guidance for the general population is not related to the topic at hand which is specifically about covid. This section can be deleted or substantially shortened. 9. Conclusions- again this section can be shortened substantially. Rather than presenting the findings once again, I suggested a simple one-two sentence synthesis of the findings. Minor points 1. P6 L17- to be precise, ergocalciferol is not made in humans as implied in this sentence, it comes from dietary sources and supplements. 2. P6 L33- add UK, so: UK guideline recommendations ... 3. P9 L33- I'm not sure whether BMJ Open requires a patient statement, but if it does, I'd suggest rewording it. I don't think that it is not possible to involve patients in this kind of research. 4. Refs 46-59 and 62 to 68 can be moved to the supplement
--	---

REVIEWER	William B. Grant Sunlight, Nutrition and Health Research Center, USA
REVIEW RETURNED	07-Sep-2020

GENERAL COMMENTS	35. Martineau AR, Jolliffe DA, Hooper RL, et al. Vitamin D supplementation to prevent acute respiratory tract infections: systematic review and meta-analysis of individual participant data. BMJ 2017;356:i6583. doi: 10.1136/bmj.i6583 This reference has apparently been updated Vitamin D supplementation to prevent acute respiratory infections: individual participant data meta-analysis. Martineau AR, Jolliffe DA, Greenberg L, Aloia JF, Bergman P, Dubnov-Raz G, Esposito S, Ganmaa D, Ginde AA, Goodall EC, Grant CC, Janssens W, Jensen ME, Kerley CP, Laaksi I, Manaseki-Holland S, Mauger D, Murdoch DR, Neale R, Rees JR, Simpson S, Stelmach I, Trilok Kumar G, Urashima M, Camargo CA, Griffiths CJ, Hooper RL. Health Technol Assess. 2019 Jan;23(2):1-44. doi: 10.3310/hta23020. 65. Meltzer DO, Best TJ, Zhang H, Vokes T, Arora V, Solway J. Association of vitamin D deficiency and treatment with COVID-19 incidence. medRxiv 2020. doi: 10.1101/2020.05.08.20095893 Is now Association of Vitamin D Status and Other Clinical Characteristics With COVID-19 Test Results. Meltzer DO, Best TJ, Zhang H, Vokes T, Arora V, Solway J. JAMA Netw Open. 2020 Sep 1;3(9):e2019722 Please check references 49 to 55 in Table 4 to see whether any more have been published in peer-reviewed journals. Additional references to consider adding Vitamin D to prevent exacerbations of COPD: systematic review and meta-analysis of individual participant data from randomised controlled trials. Jolliffe DA, Greenberg L, Hooper RL, Mathysen C, Rafiq R, de Jongh RT, Camargo CA, Griffiths CJ, Janssens W, Martineau
---

AR.Thorax. 2019 Apr;74(4):337-345. doi: 10.1136/thoraxjnl-2018-212092.

Low serum 25-hydroxyvitamin D (25[OH]D) levels in patients hospitalized with COVID-19 are associated with greater disease severity.

Panagiotou G, Tee SA, Ihsan Y, Athar W, Marchitelli G, Kelly D, Boot CS, Stock N, Macfarlane J, Martineau AR, Burns G, Quinton R. *Clin Endocrinol (Oxf)*. 2020 Jul 3:10.1111/cen.14276.

"Effect of Calcifediol Treatment and best Available Therapy versus best Available Therapy on Intensive Care Unit Admission and Mortality Among Patients Hospitalized for COVID-19: A Pilot Randomized Clinical study".

Castillo ME, Entrenas Costa LM, Vaquero Barrios JM, Alcalá Díaz JF, Miranda JL, Bouillon R, Quesada Gomez JM. *J Steroid Biochem Mol Biol*. 2020 Aug 29:105751.

Vitamin D Supplementation in COVID-19 Patients: A Clinical Case Series.

Ohaegbulam KC, Swalih M, Patel P, Smith MA, Perrin R. *Am J Ther*. 2020 Aug 13. doi: 10.1097/MJT.0000000000001222.

Vitamin D Deficiency and ARDS after SARS-CoV-2 Infection.

Faul JL, Kerley CP, Love B, O'Neill E, Cody C, Tormey W, Hutchinson K, Cormican LJ, Burke CM. *Ir Med J*. 2020 May 7;113(5):84.

Comment: excluded from analysis, but presents interesting results.

Recommend searching scholar.google.com as well. It has many more references than pubmed.gov since it does not restrict the journals included. Also, recommend extending the review to the present date since the field is very fast moving and June 10 was a long time ago.

For example, this statement is no longer correct:

"There is no peer reviewed published evidence of association between Vitamin D levels and severity of symptoms or mortality due to COVID-19."

Karonova, T. L., A. T. Andreeva and M. A. Vashukova (2020).

"[Serum 25(OH)D level in patients with CoVID-19] in Russian." *Оригинальное исследование* 12(3): 21-27.

This publication was found at scholar.google.com but not at pubmed.gov.

Rev Sanid Milit Mex 2020; 74 (1-2)

Deficiency of vitamin D is a risk factor of mortality in patients with COVID-19

Rodríguez TA, Montelongo MEA, Martínez-Cuazitl A, Puente NAV, Reyes PRA

"COVID-19 reported lower vitamin D supplementation and finally an ecological country level study that demonstrated a negative correlation between vitamin D and COVID-19 case numbers and mortality."

Comment: such ecological studies are often incorrect since the fraction of the population over the age of 70 years is a much more important factor than vitamin D status. In addition, the values used

	for vitamin D status have not been chosen to represent the values most strongly correlated with mortality rates, which are highest for the oldest people. Spurious Correlation? A review of the relationship between Vitamin D and Covid-19 infection and mortality V Kumar, A Srivastava - medRxiv, 2020 - medrxiv.org
--	--

VERSION 1 – AUTHOR RESPONSE

Reviewer:1 comments and responses

Reviewer:1 Comment 1

Can the authors please provide a table for the 4 included studies describing their characteristics? It doesn't make sense to provide a table of characteristics for the 7 excluded studies but not for the included studies.

Thank you. We have included a table of characteristics for the 4 included studies, see Table 2 of the document.

Reviewer:1 Comment 2

The Downs and Black Quality Assessment Checklist is not widely used. Could the authors please provide a brief explanation of how to interpret the scores in the risk of bias table (page 47)? What does a 14 / 20 score mean, for example?

Thank you. We have provided the following explanation, as a footnote at the bottom of table 5, in the supplemental file.

“For each included study, the maximum possible quality score was dependent on which domains could be assessed based on the study design. The higher the score assigned to a study, the lower the risk of bias. For example, Hastie et al. 2020 was assigned a score of 14 out of a maximum possible score of 20, suggesting good quality and therefore low risk of bias compared to the other studies.”

Reviewer:1 Comment 3

Please remove the “what is already known/what this study adds” sections as these are not part of journal formatting requirements.

These sections have been removed from the manuscript.

Reviewer:1 Comment 4

Please revise the ‘Strengths and limitations’ section of your manuscript (after the abstract). This section should contain five short bullet points, no longer than one sentence each, that relate specifically to the methods. The results of the study should not be summarised here.

We have amended the strengths and limitations section so that each point is shorter and relates to the methods of the systematic review and not the study findings.

Reviewer:1 Comment 5

This is a highly relevant topic as the authors point out. The methods and review itself seem to be good quality. I found the Results and Discussion long and repetitive and have made some suggestions.

Thank you. No amendment made.

Reviewer:1 Comment 6

1 P6 L45- the Martineau meta-analysis (ref 35) is used in most pieces promoting vitamin D supplementation for Covid, including this one. So, it warrants closer analysis. Some points to consider: the adjusted odds ratio of 0.88 is misleading as it overestimates the risk reduction because RTI events were common. The relative risk reduction wasn't 12%, but closer to 6%, with an absolute reduction of 2-3% which both seem pretty trivial. Currently, one subgroup analysis has been highlighted, which is quite common in the pieces promoting vitamin D. But the other subgroup analyses show the benefits were in children not adults, and those given daily or weekly but not bolus dosing of treatment. All of these subgroup analyses were largely dependent on a couple of strongly positive trials. One in particular- a cluster RCT in Mongolian children- is especially influential. But clearly generalising results from children with colds to adults with Covid, or from countries like Mongolia/Afghanistan to the UK is not appropriate. These issues should be discussed in a little more detail than is currently there. We have written about this in our editorial and responses to the Martineau paper/editorial and more recently in the Irish Medical Journal in response to ref 48. Neither of our papers need to be cited but they do cover the issue in a lot of detail which might help the authors. The authors should highlight the problem with the inappropriate generalisation of these results, which unfortunately, but predictably, has come to pass with the current covid pandemic. This issue becomes critical given the statement "If vitamin D supplementation could influence the susceptibility of people to COVID-19 infection, the intervention could be cost-effective with few associated safety risks." in p7 L7 where the Martineau results are used as the basis for suggesting vitamin D supplementation might be beneficial for covid. In my view, the Martineau results don't suggest any likely benefit once they examined at anything more than a superficial level.

We appreciate the critique of the Martineau paper, however the results provided in our study replicate those in the BMJ publication (and the subsequent HTA monograph). We have added the following sentence to demonstrate the uncertainty you have highlighted in the BMJ editorial: "Critiques of this review have suggested that the findings should be interpreted as hypothesis generating only as the results are heterogeneous and not sufficiently applicable to the general population" on page 5-6.

Due to the uncertainty highlighted in the paragraphs above and the editorial we have removed the following sentence from the introduction "If vitamin D supplementation could influence the susceptibility of people to COVID-19 infection, the intervention could be cost-effective with few associated safety risks." Unfortunately, we were unable to find your responses to the Martineau BMJ paper from those listed on the BMJ website, however we were able to find the editorial which we have included in the references.

Reviewer:1 Comment 7

2. Since policy about vitamin D supplements and covid will be best informed by RCTs, the Results should start with an explicit statement that no relevant RCTs were identified e.g. somewhere in the first paragraph, and this should also be mentioned explicitly in the Discussion and Abstract

We have included a statement that no RCTs were identified or included in the first paragraph of the results section. We have also made this explicit in the Discussion and Abstract of the manuscript.

Reviewer:1 Comment 8

3. Much of the narrative detail in the Results can be presented much more simply in a Table similar to the table of study characteristics in the supplement. This would allow the key points related to vitamin

D and covid only to be presented in the text, and the Results section could then be much briefer and more focussed.

The addition of a table of characteristics for the 4 included studies has been made in response to comment 1. We have removed some of the descriptive text regarding study characteristics which have been summarised in Table 2.

Reviewer:1 Comment 9

4. The data from the studies is incredibly weak as the authors discuss, but I think that needs to be highlighted more. The UK biobank study showed no effect. The Swiss study didn't present sufficient data to know whether the groups were comparable. The Italian case-control simply records vitamin D supplement usage, not even 25OHD levels. The ecological study is largely worthless as it only compares vitamin D levels and covid rates at a country level when obviously far more important factors are not considered. Rather than repeating the results in the first paragraph of the Discussion, I suggest the authors synthesize them. My interpretation of the Results is that there is no robust evidence of any link between 25OHD and covid.

We have included additional text which describes the poor quality of the studies in the results and discussion sections of the manuscript for each included study pg. 11-12. We have reduced the results mentioned in the discussion section as requested in other reviewer 1 comments.

Reviewer:1 Comment 10

5. Paragraphs 2-4 of the Discussion are largely redundant because the findings are either neutral or not robust. This section can be shortened substantially or deleted.

We have removed/shortened and combined a lot of the detail presented in paragraphs 2-4 of the manuscript discussion.

Reviewer:1 Comment 11

6. P13, L33 – there is no evidence, not “limited” evidence isn't there? Likewise in the Abstract conclusions: the first sentence differs from the second and seems both unnecessary and inconsistent with the actual findings of no reliable evidence rather than “limited” evidence.

We have amended limited to “no robust evidence” throughout the manuscript and adjusted the first two sentences of the abstract conclusion to limit confusion. They now read as follows: “There is no robust evidence of a negative association between vitamin D and COVID-19 infection. No relevant randomised control trials were identified and there are no peer reviewed published evidence of association between Vitamin D levels and severity of symptoms or mortality due to COVID-19.”

Reviewer:1 Comment 12

7. P13, L44-52 – I don't understand the relevance of this text, or why speculation about vitamin D's effects is appropriate in the absence of evidence of effect.

We wanted to address the speculation evident in the literature and media surrounding vit D and COVID-19 in our paper. We have amended this paragraph to include the following statement: “However, our systematic review found no robust evidence of a negative association between vitamin D and COVID-19 infection. Publication of peer reviewed research specific to COVID-19 infection is required to identify and validate exact mechanisms involved in the human population”.

Reviewer:1 Comment 13

8. Implications for practice section- presenting the guidance for the general population is not related to the topic at hand which is specifically about covid. This section can be deleted or substantially shortened.

We have shorted this section on pg. 13 but retained the citations.

Reviewer:1 Comment 14

9. Conclusions- again this section can be shortened substantially. Rather than presenting the findings once again, I suggested a simple one-two sentence synthesis of the findings.

We have largely removed the study results from the conclusion, thereby shortening this section.

Reviewer:1 Comment 15

1. P6 L17- to be precise, ergocalciferol is not made in humans as implied in this sentence, it comes from dietary sources and supplements.

Thank you, for accuracy we have corrected this statement on page 5.

Reviewer:1 Comment 16

2. P6 L33- add UK, so: UK guideline recommendations ...

We have added UK on page 5.

Reviewer:1 Comment 17

3. P9 L33- I'm not sure whether BMJ Open requires a patient statement, but if it does, I'd suggest rewording it. I don't think that it is not possible to involve patients in this kind of research.

We agree with this comment and have reworded the PPI statement on page 8 as follows: "Due to the rapid timeframe of this systematic review it was not possible for our research team to involve patients or the public in the design, conduct, or reporting of our study."

Reviewer:1 Comment 18

4. Refs 46-59 and 62 to 68 can be moved to the supplement

We believe that references 46-59 (now 49-62) should remain in the manuscript for accuracy as they were reviewed as part of the search strategy/screening process. We cite references 62-68 (now 63-69) in the text as the 'seven studies were excluded because they are not available as peer reviewed publications'. We would be happy to move 62 to 68 (now 63-69) to the supplement if the editorial team felt this was required, however we consider the citations 62-68 (now 63-69) to be in the appropriate place.

Reviewer:2 comments and responses

Reviewer:2 Comment 19

35. Martineau AR, Jolliffe DA, Hooper RL, et al. Vitamin D supplementation to prevent acute respiratory tract infections: systematic review and meta-analysis of individual participant data. *BMJ* 2017;356:i6583. doi: 10.1136/bmj.i6583

This reference has apparently been updated: Vitamin D supplementation to prevent acute respiratory infections: individual participant data meta-analysis.

Martineau AR, Jolliffe DA, Greenberg L, Aloia JF, Bergman P, Dubnov-Raz G, Esposito S, Ganmaa D, Ginde AA, Goodall EC, Grant CC, Janssens W, Jensen ME, Kerley CP, Laaksi I, Manaseki-Holland S, Mauger D, Murdoch DR, Neale R, Rees JR, Simpson S, Stelmach I, Trilok Kumar G, Urashima M, Camargo CA, Griffiths CJ, Hooper RL. *Health Technol Assess*. 2019 Jan;23(2):1-44. doi: 10.3310/hta23020.

Thank you, we have checked this update and it appears to be the same study in two different journals, the HTA monograph and the BMJ article. The abstracts, particularly the results section, are almost identical as they refer to the same study. It is our preference to cite the BMJ paper as it was available first.

Reviewer:2 Comment 20

65. Meltzer DO, Best TJ, Zhang H, Vokes T, Arora V, Solway J. Association of vitamin D deficiency and treatment with COVID-19 incidence. *medRxiv* 2020. doi: 10.1101/2020.05.08.20095893
Is now Association of Vitamin D Status and Other Clinical Characteristics With COVID-19 Test Results.

Meltzer DO, Best TJ, Zhang H, Vokes T, Arora V, Solway J. *JAMA Netw Open*. 2020 Sep 1;3(9):e2019722

We note that this is the same study which has been updated for the JAMA Open publication (note the 10 fewer patients in the analysis). The new publication is available at <https://jamanetwork.com/journals/jamanetworkopen/fullarticle/2770157>

The version accessed during our search was the pre-print and can be found here <https://www.medrxiv.org/content/10.1101/2020.05.08.20095893v1>

However, the JAMA Open publication was published September 3rd 2020, and therefore, after the search conducted for our systematic review.

To maintain the accuracy of our search, and to ensure that our whole paper reflects the literature available at time of last search/synthesis in June this citation cannot be updated in the reference list – as stated in this sentence “Seven articles met the eligibility criteria but were excluded as they were not available as peer reviewed publications at the time of our narrative synthesis”

The JAMA Open publication would be found in a future update search using our search strategy which we included in the supplement.

Reviewer:2 Comment 21

Please check references 49 to 55 in Table 4 to see whether any more have been published in peer-reviewed journals.

In line with our response to comment 20, checking/updating these references will not alter the studies that are included in this systematic review as it risks the integrity of the record of the literature at time of our systematic search and synthesis. However, we have added an acknowledgement in the supplementary appendix Table 4 to signify to the reader that two of the seven papers are available as peer reviewed publications since the systematic review was performed. As before, we have provided citations and links to the studies for the reader.

Reviewer:2 Comment 22

Additional references to consider adding

1. Vitamin D to prevent exacerbations of COPD: systematic review and meta-analysis of individual participant data from randomised controlled trials.

Jolliffe DA, Greenberg L, Hooper RL, Mathysen C, Rafiq R, de Jongh RT, Camargo CA, Griffiths CJ, Janssens W, Martineau AR. *Thorax*. 2019 Apr;74(4):337-345. doi: 10.1136/thoraxjnl-2018-212092.

2. Low serum 25-hydroxyvitamin D (25[OH]D) levels in patients hospitalized with COVID-19 are associated with greater disease severity.

Panagiotou G, Tee SA, Ihsan Y, Athar W, Marchitelli G, Kelly D, Boot CS, Stock N, Macfarlane J, Martineau AR, Burns G, Quinton R. *Clin Endocrinol (Oxf)*. 2020 Jul 3:10.1111/cen.14276.

3. "Effect of Calcifediol Treatment and best Available Therapy versus best Available Therapy on Intensive Care Unit Admission and Mortality Among Patients Hospitalized for COVID-19: A Pilot Randomized Clinical study".

Castillo ME, Entrenas Costa LM, Vaquero Barrios JM, Alcalá Díaz JF, Miranda JL, Bouillon R, Quesada Gomez JM. *J Steroid Biochem Mol Biol*. 2020 Aug 29:105751.

4. Vitamin D Supplementation in COVID-19 Patients: A Clinical Case Series.

Ohaegbulam KC, Swalih M, Patel P, Smith MA, Perrin R. *Am J Ther*. 2020 Aug 13. doi: 10.1097/MJT.0000000000001222.

5. Vitamin D Deficiency and ARDS after SARS-CoV-2 Infection.

Faul JL, Kerley CP, Love B, O'Neill E, Cody C, Tormey W, Hutchinson K, Cormican LJ, Burke CM. *Ir Med J*. 2020 May 7;113(5):84.

Comment: excluded from analysis, but presents interesting results.

Recommend searching scholar.google.com as well. It has many more references than pubmed.gov since it does not restrict the journals included.

We have numbered the citations provided above, our information specialist has scrutinised citations 1-5. Citation 1 was not returned in our search because the condition is COPD not coronaviruses. Citations 2-4 were not included in our review because they were published after our systematic review was conducted. Finally, citation 5 was returned in our search but excluded during the screening process.

We did not include scholar.google.com in this review, and therefore it was not listed as a database in our detailed search strategy (provided in the supplement). According to the Cochrane Handbook technical supplement the precise number of journals indexed by Google Scholar is not known because it does not use a pre-specified list of journals to populate its content. A further disadvantage of Google Scholar is that the export features are basic and inefficient and are only marginally improved by linking to its preferred reference manager software, Zotero (Bramer et al 2013, Levay et al 2016). Finally, Google Scholar limits the number of viewable results to 1000 and does not disclose how the top 1000 results are selected, thus compromising the transparency and reproducibility of search results (Levay et al 2016). We attempted to capture recent and upcoming publications in our search via inclusion of various living systematic reviews on COVID-19.

Reviewer:2 Comment 23

Also, recommend extending the review to the present date since the field is very fast moving and June 10 was a long time ago.

We agree that the field is very fast moving, hence we updated the search prior to submission of the paper for publication to enable us to capture any recent publications (10-May, update 10-June). The search strategy is less than 6 months - 1 year old which is in line with Cochrane guidelines for publications. Given this is a rapid response with a fairly recent search strategy and limited resources we are unable to update searches. The full search strategy is included in the supplement for transparency and future updates. We note in the discussion that this is a fast growing field and updates should be considered as a research priority.

Reviewer:2 Comment 24

For example, this statement is no longer correct:

“There is no peer reviewed published evidence of association between Vitamin D levels and severity of symptoms or mortality due to COVID-19.”

Karonova, T. L., A. T. Andreeva and M. A. Vashukova (2020). "[Serum 25(OH)D level in patients with CoVID-19] in Russian." Оригинальное исследование 12(3): 21-27.

In line with our response to comments 20 and 21, this citation record was first created in Embase on 28/8/20 which was after our search was completed. However, it would be returned in a future update search using the search strategy available in our supplement. To retain the accuracy of our systematic review at the time in which it was performed we believe that the core text should not be amended.

Reviewer:2 Comment 25

This publication was found at scholar.google.com but not at pubmed.gov.

Rev Sanid Milit Mex 2020; 74 (1-2) Deficiency of vitamin D is a risk factor of mortality in patients with COVID-19 Rodríguez TA, Montelongo MEA, Martínez-Cuazitl A, Puente NAV, Reyes PRA

“COVID-19 reported lower vitamin D supplementation and finally an ecological country level study that demonstrated a negative correlation between vitamin D and COVID-19 case numbers and mortality.”

Comment: such ecological studies are often incorrect since the fraction of the population over the age of 70 years is a much more important factor than vitamin D status. In addition, the values used for vitamin D status have not been chosen to represent the values most strongly correlated with mortality rates, which are highest for the oldest people.

Our information specialist has searched for this citation record in the Medline (which does have some articles from this journal), Embase, and the Bern, Oxford & WHO Covid databases and it is not available. We were able to identify it on the journal website for ‘Revista de Sandidad Militar’

<https://www.medigraphic.com/cgi-bin/new/resumenI.cgi?IDARTICULO=93773>

The journal issue is January-April 2020, we suggest that the paper was not made available online until recently and that’s why none of the databases have picked it up yet. Nevertheless, we found this statement in the webpage source code: which clearly demonstrates that it was published after our final search date of 10th June 2020.

Reviewer:2 Comment 26

Spurious Correlation? A review of the relationship between Vitamin D and Covid-19 infection and mortality

V Kumar, A Srivastava - medRxiv, 2020 - medrxiv.org

This citation was returned in our search but excluded during the screening process, it is an unpublished review (i.e., not peer reviewed).

VERSION 2 – REVIEW

REVIEWER	Bolland, Mark University of Auckland, Department of Medicine
REVIEW RETURNED	04-Nov-2020

GENERAL COMMENTS	The authors have responded reasonably to the comments I made previously.
--

REVIEWER	William B. Grant Sunlight, Nutrition and Health Research Center USA
REVIEW RETURNED	01-Nov-2020

GENERAL COMMENTS	We searched the global research on COVID-19 developed by the WHO,⁴⁴ CEBM Oxford,⁴⁵ and the living systematic review developed by Bern University⁴⁶ on 10 May 2020. We updated the database searches on 10th June 2020 to capture articles which may have been published since the initial search was conducted Comment: It is now 31 October, and there are many more observational and intervention studies re 25(OH)D and vitamin D supplementation than included in the manuscript. Any reader would like to know the situation near the time of publication of this analysis, not what it was like 3.7 months ago. See this recent review for studies to consider including in the analysis. Mercola, J.; Grant, W.B.; Wagner, C.L. Evidence Regarding Vitamin D and Risk of COVID-19 and Its Severity. Nutrients 2020, 12, 3361. doi:10.3390/nu12113361 As well as this study published after that one. Possible association of vitamin D status with lung involvement and outcome in patients with COVID-19: a retrospective study. Abrishami A, Dalili N, Mohammadi Torbati P, Asgari R, Arab-Ahmadi M, Behnam B, Sanei-Taheeri M. Eur J Nutr. 2020 Oct 30. doi: 10.1007/s00394-020-02411-0. There are now two Hastie et al. papers on UK Biobank data. They should be considered inappropriate for two reasons: 1, the 25(OH)D values are too old; 2, the adjusted odds ratios included factors that affect 25(OH)D. Response to 'Vitamin D concentrations and COVID-19 infection in UK Biobank'. Roy AS, Matson M, Herlekar R. Diabetes Metab Syndr. 2020 Sep-Oct;14(5):777. doi: 10.1016/j.dsx.2020.05.049. Principles of confounder selection. VanderWeele TJ. Eur J Epidemiol. 2019 Mar;34(3):211-219. doi: 10.1007/s10654-019-00494-6.
--

	Non-peer reviewed preprints should not be included in the analysis as they may be incorrect. Studies that use population-mean 25OHD levels or latitude are not really measuring 25OHD for those who develop COVID-19 so should not be included. The population that develops severe COVID-19 tends to be elderly, obese, with comorbid conditions, in nursing homes, and/or of lower socioeconomic status, etc., and would not have 25OHD levels of the general population. Conclusion: There is no robust evidence of a negative association between vitamin D and COVID-19 infection. and there is no robust peer reviewed published evidence of association between Vitamin D levels and severity of symptoms or mortality due to COVID-19. Guideline producers should acknowledge that benefits of vitamin D supplementation in COVID-19 infection are as yet unproven despite increasing interest from the media and academic community Comment: This statement is no longer correct. No relevant randomised control trials were identified Comment: This statement is correct but does not override the findings from observational studies.
--	--

VERSION 2 – AUTHOR RESPONSE

Response to reviewers' comments:

Reviewer: 1

Dr. Mark Bolland, University of Auckland

We thank reviewer 1 who confirmed that we responded reasonably to their comments.

Reviewer: 2

Mr. William Grant, Sunlight, Nutrition, and Health Research Center

Reviewer:2 Comment 1

We searched the global research on COVID-19 developed by the WHO,⁴⁴ CEBM Oxford,⁴⁵ and the living systematic review developed by Bern University⁴⁶ on 10 May 2020. We updated the database searches on 10th June 2020 to capture articles which may have been published since the initial search was conducted. It is now 31 October, and there are many more observational and intervention studies re 25(OH)D and vitamin D supplementation than included in the manuscript. Any reader would

like to know the situation near the time of publication of this analysis, not what it was like 3.7 months ago.

We appreciate this statement. However, the time between our systematic review search date and our initial submission to manuscript central was less than 1 calendar month. Reviewer 2 and the editorial team should acknowledge that the delay in publication since the searches were conducted (10th June) is a combination of both the standard review process (first submitted to the BMJ 6th July, returned 24th July, transferred to BMJ Open 5th Aug, reviewer comments were received 2th Oct) and the significant delays in the process due to COVID-19 – as outlined in the COVID-19 statement from BMJ Author Hub, <https://authors.bmj.com/policies/covid-19/>.

In addition to our emails to manuscript central requesting progress updates, we responded to the first set of reviewer comments within 16 working days (submitted 27th Oct, returned 22nd Dec). During the period between 6th July to 22nd Dec (6 months inclusive) we appreciate that there are inevitably more studies regarding 25(OH)D and vitamin D supplementation than included in our manuscript. We would like to reiterate our original response to this point which we made in the first round of reviewer comments to reviewer 2 (Reviewer:2 Comment 23), which stated that:

'We agree that the field is very fast moving, hence we updated the search prior to submission of the paper for publication to enable us to capture any recent publications (10-May, update 10-June). The search strategy is less than 6 months - 1 year old which is in line with Cochrane guidelines for publications. Given this is a rapid response with a fairly recent search strategy and limited resources we are unable to update searches. The full search strategy is included in the supplement for transparency and future updates. We note in the discussion that this is a fast-growing field and updates should be considered as a research priority.'

Reviewer:2 Comment 2

See this recent review for studies to consider including in the analysis.

Mercola, J.; Grant, W.B.; Wagner, C.L. Evidence Regarding Vitamin D and Risk of COVID-19 and Its Severity. *Nutrients* 2020, 12, 3361. doi:10.3390/nu12113361

We note that this citation is a non-systematic review (therefore, would not be included in our systematic review) which was published: 31 October 2020, which is after our June search date. This paper includes 13 or 14 observational studies, we note that the manuscript text states 14 studies but only 13 studies are listed in table.

As per our first set of reviewer comment responses (Reviewer:2 Comment 22) we have considered the studies as suggested by reviewer 2. We would like to note that the aim of the systematic review method is to systematically consider all relevant literature using a clearly formulated question, and systematic and reproducible methods to identify, select and critically appraise all relevant research,

and to collect and analyse data from the studies that are included in the review. We conducted a systematic review so as to not review literature in an ad hoc manner, nevertheless we have commented on 13 studies included in Mercola et al 2020 below:

1. Hastie, C.E.; Mackay, D.F.; Ho, F.; Celis-Morales, C.A.; Katikireddi, S.V.; Niedzwiedz, C.L.; Jani, B.D.; Welsh, P.; Mair, F.S.; Gray, S.R.; et al. Vitamin D concentrations and COVID-19 infection in UK Biobank. *Diabetes Metab. Syndr. Clin. Res. Rev.* 2020, 14, 561–565. [Google Scholar] [CrossRef]

This is included in our review.

2. D'Avolio, A.; Avataneo, V.; Manca, A.; Cusato, J.; De Nicolo, A.; Lucchini, R.; Keller, F.; Cantu, M. 25-hydroxyvitamin d concentrations are lower in patients with positive pcr for sars-cov-2. *Nutrients* 2020, 12, 1359. [Google Scholar] [CrossRef] [PubMed]

This is included in our review.

3. Panagiotou, G.; Tee, S.A.; Ihsan, Y.; Athar, W.; Marchitelli, G.; Kelly, D.; Boot, C.S.; Stock, N.; Macfarlane, J.; Martineau, A.R.; et al. Low serum 25-hydroxyvitamin D (25[OH]D) levels in patients hospitalised with COVID-19 are associated with greater disease severity: Results of a local audit of practice. *Clin. Endocrinol. (Oxf.)* 2020. [Google Scholar] [CrossRef]

This is a medRxiv preprint first available 3rd July 2020 and it is not published in a peer reviewed journal, therefore it would not be included in our systematic review or results. If we were to update our search, we will include it in our separate supplemental file's table as a medRxiv preprint.

4. Carpagnano, G.E.; Di Lecce, V.; Quaranta, V.N.; Zito, A.; Buonamico, E.; Capozza, E.; Palumbo, A.; Di Gioia, G.; Valerio, V.N.; Resta, O. Vitamin D deficiency as a predictor of poor prognosis in patients with acute respiratory failure due to COVID-19. *J. Endocrinol. Investig.* 2020, 1–7. [Google Scholar] [CrossRef]

This study was published: 09 August 2020, therefore if we are to conduct an update to our search it would be reviewed as part of the systematic review process and may be included in the final results if appropriate.

5. Im, J.H.; Je, Y.S.; Baek, J.; Chung, M.H.; Kwon, H.Y.; Lee, J.S. Nutritional status of patients with coronavirus disease 2019 (covid-19). *Int. J. Infect. Dis.* 2020, 100, 390–393. [Google Scholar] [CrossRef] [PubMed]

This study was published: 11 August 2020, therefore if we are to conduct an update to our search it would be reviewed as part of the systematic review process and may be included in the final results if appropriate.

6. Karonova, T.L.; Andreeva, A.T.; Vashukova, M.A. Serum 25(OH)D Level in Patients with Covid-19. *J. Infectol.* 2020, 12, 21–27. (In Russian) [Google Scholar] [CrossRef]

This is a Russian publication published: August 28th 2020, which would not be included in our systematic review due to 'language'.

7. Pérez, R.A.R.; Nieto, A.V.P.; Martínez-Cuazitl, A.; Mercado, E.A.M.; Tort, A.R. La deficiencia de vitamina D es un factor de riesgo de mortalidad en pacientes con COVID-19. *Rev. Sanid. Mil.* 2020, 74, 106–113. [Google Scholar] [CrossRef]

We have already provided our response regarding this paper in the first set of reviewer comments (see Reviewer:2 Comment 25). We note that this citation is

referenced wrongly here; main author is Rodriguez Tort, not Perez. During the first review this paper was not available on Medline, Embase, and the Bern, Oxford & WHO Covid databases, and we have checked again 6th Jan 2021 it is still not in Medline, Pubmed, Embase, nor the Oxford, Bern or WHO databases. It's only findable via Google/Google Scholar. Nevertheless this is a Spanish publication which would not be included in our systematic review due to 'language'.

8. Baktash, V.; Hosack, T.; Patel, N.; Shah, S.; Kandiah, P.; Abbeele, K.V.D.; Mandal, A.K.J.; Missouriis, C.G. Vitamin D status and outcomes for hospitalised older patients with COVID-19 2020. *Postgrad. Med. J.* 2020. [Google Scholar] [CrossRef]

This study was published: August 27, 2020., therefore if we are to conduct an update to our search it would be reviewed as part of the systematic review process and may be included in the final results, however we expect not as it only mentions: COVID-19-positive arm demonstrated lower median serum 25(OH)D level of 27 nmol/L (IQR=20–47 nmol/L) compared with COVID-19-negative arm, with median level of 52 nmol/L (IQR=31.5–71.5 nmol/L) (p value=0.0008).

9. Hastie, C.E.; Pell, J.P.; Sattar, N. Vitamin D and COVID-19 infection and mortality in UK Biobank. *Eur. J. Nutr.* 2020, 1–4. [Google Scholar] [CrossRef] [PubMed]

This study was published: 26 August 2020, therefore if we are to conduct an update to our search it would be reviewed as part of the systematic review process and may be included in the final results if appropriate.

10. Radujkovic, A.; Hippchen, T.; Tiwari-Heckler, S.; Dreher, S.; Boxberger, M.; Merle, U. Vitamin D Deficiency and Outcome of COVID-19 Patients. *Nutrients* 2020, 12, 2757. [Google Scholar] [CrossRef]

This study was published: 10 Sept 2020, therefore if we are to conduct an update to our search it would be reviewed as part of the systematic review process and may be included in the final results if appropriate.

11. Pizzini, A.; Aichner, M.; Sahanic, S.; Böhm, A.; Egger, A.; Hoermann, G.; Kurz, K.; Widmann, G.; Bellmann-Weiler, R.; Weiss, G.; et al. Impact of Vitamin D Deficiency on COVID-19—A Prospective Analysis from the CovILD Registry. *Nutrients* 2020, 12, 2775. [Google Scholar] [CrossRef] [PubMed]

This study was published: 11 Sept 2020, therefore if we are to conduct an update to our search it would be reviewed as part of the systematic review process and may be included in the final results if appropriate.

12. Macaya, F.; Paeres, C.E.; Valls, A.; Fernández-Ortiz, A.; Del Castillo, J.G.; Martín-Sánchez, J.; Runkle, I.; Herrera, M.; Ángel, R. Interaction between age and vitamin D deficiency in severe COVID-19 infection. *Nutrición Hospitalaria* 2020. [Google Scholar] [CrossRef] Accepted for publication: 20/07/2020, It is in Spanish: abstract is in English. So, after updating we will not include it! I did not even find a pdf (Anna confirmed it).

We were unable to find the full version of this article. However, we note that this is a Spanish publication which would not be included in our systematic review due to 'language'.

13. Ye, K.; Tang, F.; Liao, X.; Shaw, B.A.; Deng, M.; Huang, G.; Qin, Z.; Peng, X.; Xiao, H.; Chen, C.; et al. Does Serum Vitamin D Level Affect COVID-19 Infection and Its Severity?-A Case-Control Study. *J. Am. Coll. Nutr.* 2020, 1–8. [Google Scholar] [CrossRef] [PubMed]

This study was published: 13 Oct 2020, therefore if we are to conduct an update to our search it would be reviewed as part of the systematic review process and may be included in the final results if appropriate.

Reviewer:2 Comment 3

As well as this study published after that one.

Possible association of vitamin D status with lung involvement and outcome in patients with COVID-19: a retrospective study.

Abrishami A, Dalili N, Mohammadi Torbati P, Asgari R, Arab-Ahmadi M, Behnam B, Sanei-Taheri M. *Eur J Nutr.* 2020 Oct 30. doi: 10.1007/s00394-020-02411-0.

See our initial response to Reviewer:2 Comment 2. We conducted a systematic review so as to not review literature in an ad hoc manner, nevertheless we have commented on the study mentioned by reviewer 2 below:

1. Abrishami A, Dalili N, Mohammadi Torbati P, Asgari R, Arab-Ahmadi M, Behnam B, Sanei-Taheri M. Possible association of vitamin D status with lung involvement and outcome in patients with COVID-19: a retrospective study. *M.Eur J Nutr.* 2020 Oct 30. doi: 10.1007/s00394-020-02411-0.

This study was published: 30 Oct 2020, therefore if we are to conduct an update to our search it would be reviewed as part of the systematic review process and may be included in the final results if appropriate.

Reviewer:2 Comment 4

There are now two Hastie et al. papers on UK Biobank data. They should be considered inappropriate for two reasons: 1, the 25(OH)D values are too old; 2, the adjusted odds ratios included factors that affect 25(OH)D.

We note that one of the Hastie papers is included in our systematic review (citation 1

Reviewer:2 Comment 2), the second Hastie paper has been addressed in citation 9 as part of Reviewer:2 Comment 2.

We do not agree with the reviewer's opinion that the Hastie studies should be considered 'inappropriate' without full assessment against our systematic review inclusion criteria. The first paper met our inclusion criteria, therefore was included in our systemic review, the methodological weaknesses of the study are described in our manuscript.

Reviewer:2 Comment 5

Response to 'Vitamin D concentrations and COVID-19 infection in UK Biobank'.

Roy AS, Matson M, Herlekar R. *Diabetes Metab Syndr*. 2020 Sep-Oct;14(5):777. doi: 10.1016/j.dsx.2020.05.049.

We note that this study was found in our search in June. It is a letter/comment on Hastie that would not be included in our updated systematic review due to 'study design'.

Reviewer:2 Comment 6

Principles of confounder selection.

VanderWeele TJ. *Eur J Epidemiol*. 2019 Mar;34(3):211-219. doi: 10.1007/s10654-019-00494-6.

Thank you for providing this citation.

Reviewer:2 Comment 7

Non-peer reviewed preprints should not be included in the analysis as they may be incorrect.

We agree with this statement. Non-peer reviewed preprints were not included in our systematic review. We point the reviewer to comment 2 above, where we are asked to consider non-peer reviewed preprints as part of reviewer 2's own review article: Mercola, J.; Grant, W.B.; Wagner, C.L. Evidence Regarding Vitamin D and Risk of COVID-19 and Its Severity. *Nutrients* 2020, 12, 3361. doi:10.3390/nu12113361 and to the reviewers first set of comments (Reviewer:2 Comment 22 and Reviewer:2 Comment 26) which also included non-peer reviewed preprints which we were asked to consider.

Reviewer:2 Comment 7

Studies that use population-mean 25OHD levels or latitude are not really measuring 25OHD for those who develop COVID-19 so should not be included. The population that develops severe COVID-19 tends to be elderly, obese, with comorbid conditions, in nursing homes, and/or of lower socioeconomic status, etc., and would not have 25OHD levels of the general population.

Thank you for providing this commentary. No response required.

Reviewer:2 Comment 8

Conclusion: There is no robust evidence of a negative association between vitamin D and COVID-19 infection. and there is no robust peer reviewed published evidence of association between Vitamin D levels and severity of symptoms or mortality due to COVID-19. Guideline producers should acknowledge that benefits of vitamin D supplementation in COVID-19 infection are as yet unproven despite increasing interest from the media and academic community Comment: This statement is no longer correct.

The statement copied above is factually correct in accordance with our systematic review which was included in the manuscript.

Reviewer:2 Comment 9

No relevant randomised control trials were identified

Comment: This statement is correct but does not override the findings from observational studies.

Thank you. No response required.

VERSION 3 – REVIEW

REVIEWER	Grant, William Sunlight, Nutrition, and Health Research Center
REVIEW RETURNED	09-Jan-2021

GENERAL COMMENTS	This manuscript is a systematic review of publications regarding vitamin D supplementation and clinical outcome for COVID-19. Unfortunately, the literature search was last conducted on June 10, 2020, so recent studies that should be included in the review were not included. Title: COVID-19 infection is incorrect. It is either SARS-CoV-2 infection or COVID-19 (the ensuing disease). It should be made clear whether disease or disease and infection are the subjects of this review. Evidently it is both as in Table 2, D'Avolio looks at infection, Hastie looks at COVID-19, Ilie and Fasano look at COVID-19. Also, vitamin D supplementation or level is incorrect; should be vitamin D supplementation or serum 25-hydroxyvitamin D level. Most recently, Martineau and colleagues (2017) conducted a systematic review and meta-analysis of individual participant data from randomised controlled trials (RCTs) to assess the overall effect of vitamin D supplementation on risk of acute RTI.³⁸ Comment: There is a more recent version of this work: Vitamin D supplementation to prevent acute respiratory infections: individual participant data meta-analysis. Martineau AR, Jolliffe DA, Greenberg L, Aloia JF, Bergman P, Dubnov-Raz G, Esposito S, Ganmaa D, Ginde AA, Goodall EC, Grant CC, Janssens W, Jensen ME, Kerley CP, Laaksi I,
---

Manaseki-Holland S, Mauger D, Murdoch DR, Neale R, Rees JR, Simpson S, Stelmach I, Trilok Kumar G, Urashima M, Camargo CA, Griffiths CJ, Hooper RL. *Health Technol Assess.* 2019 Jan;23(2):1-44. doi: 10.3310/hta23020.

Recent rapid reviews of vitamin D for treatment or prevention in COVID-19 reported no evidence that vitamin D deficiency predisposes to COVID-19, or that vitamin D supplementation is effective in prevention or treatment of COVID-19.^{40 41} However, data sources included in the rapid review were limited.⁴²

Comment: Refs. 40 and 41 are from June and are too old now; Ref. 42 is from May.

The search strategy was developed by the information specialists in collaboration with the research team and clinical advisors. We searched MEDLINE (OVID interface), Embase (OVID interface), Cochrane Central Register of Controlled Trials, MedRxiv and BioRxiv preprint databases on 6th-8th May 2020. We searched the global research on COVID-19 developed by the WHO,⁴⁴ CEBM Oxford,⁴⁵ and the living systematic review developed by Bern University⁴⁶ on 10 May 2020. We updated the database searches on 10th June 2020 to capture articles which may have been published since the initial search was conducted. Comment: This is a very fast moving field of research. 10 June is too long ago; the search must be updated in January 2021. Preprint servers should not be used if they have only non-peer reviewed preprints. Scholar.google.com should be used in addition to pubmed.gov.

Here are some of the new articles to include:

Association of Vitamin D Status and Other Clinical Characteristics With COVID-19 Test Results.

Meltzer DO, Best TJ, Zhang H, Vokes T, Arora V, Solway J. *JAMA Netw Open.* 2020 Sep 1;3(9):e2019722. doi: 10.1001/jamanetworkopen.2020.19722.

Low plasma 25(OH) vitamin D level is associated with increased risk of COVID-19 infection: an Israeli population-based study.

Merzon E, Tworowski D, Gorohovski A, Vinker S, Golan Cohen A, Green I, Frenkel-Morgenstern M. *FEBS J.* 2020 Sep;287(17):3693-3702. doi: 10.1111/febs.15495. Epub 2020 Aug 28.

"Effect of calcifediol treatment and best available therapy versus best available therapy on intensive care unit admission and mortality among patients hospitalized for COVID-19: A pilot randomized clinical study".

Entrenas Castillo M, Entrenas Costa LM, Vaquero Barrios JM, Alcalá Díaz JF, López Miranda J, Bouillon R, Quesada Gomez JM. *J Steroid Biochem Mol Biol.* 2020 Oct;203:105751. doi: 10.1016/j.jsbmb.2020.105751.

Vitamin D and survival in COVID-19 patients: A quasi-experimental study.

	Annweiler C, Hanotte B, Grandin de l'Eprevier C, Sabatier JM, Lafaie L, C�larier T.J Steroid Biochem Mol Biol. 2020 Nov;204:105771. doi: 10.1016/j.jsbmb.2020.105771 Vitamin D Supplementation Associated to Better Survival in Hospitalized Frail Elderly COVID-19 Patients: The GERIA-COVID Quasi-Experimental Study. Annweiler G, Corvaisier M, Gautier J, Dub�e V, Legrand E, Sacco G, Annweiler C.Nutrients. 2020 Nov 2;12(11):3377. doi: 10.3390/nu12113377. SARS-CoV-2 positivity rates associated with circulating 25-hydroxyvitamin D levels. Kaufman HW, Niles JK, Kroll MH, Bi C, Holick MF.PLoS One. 2020 Sep 17;15(9):e0239252. doi: 10.1371/journal.pone.0239252. eCollection 2020. Mortality in an Italian nursing home during COVID-19 pandemic: correlation with gender, age, ADL, vitamin D supplementation, and limitations of the diagnostic tests. Cangiano B, Fatti LM, Danesi L, Gazzano G, Croci M, Vitale G, Gilardini L, Bonadonna S, Chiodini I, Caparello CF, Conti A, Persani L, Stramba-Badiale M, Bonomi M.Aging (Albany NY). 2020 Dec 22;12(24):24522-24534. Vitamin D supplementation and outcomes in coronavirus disease 2019 (COVID-19) patients from the outbreak area of Lombardy, Italy. Cereda E, Bogliolo L, Lobascio F, Barichella M, Zecchinelli AL, Pezzoli G, Caccialanza R.Nutrition. 2020 Nov 11:111055. doi: 10.1016/j.nut.2020.111055.
--	---

VERSION 3 – AUTHOR RESPONSE

Response to reviewers' comments:

Comment 1. This manuscript is a systematic review of publications regarding vitamin D supplementation and clinical outcome for COVID-19. Unfortunately, the literature search was last conducted on June 10, 2020, so recent studies that should be included in the review were not included.

Response: The authors acknowledge that this review was conducted on 10th June 2020. Recent studies published will be screening and included where relevant in our forthcoming update of this systematic review which will be published at a later date.

Comment 2. Title: COVID-19 infection is incorrect. It is either SARS-CoV-2 infection or COVID-19 (the ensuing disease). It should be made clear whether disease or disease and infection are the subjects of this review. Evidently it is both as in Table 2, D'Avolio looks at infection, Hastie looks at COVID-19, Ilie and Fasano look at COVID-19. Also, vitamin D supplementation or level is incorrect; should be vitamin D supplementation or serum 25-hydroxyvitamin D level.

Response: The title has been amended as follows: We have also changed the text through out to maintain consistency with the title.

Objective: To systemically review and critically appraise published studies of the association between vitamin D supplementation or serum vitamin D level and susceptibility to SARS-CoV-2 infection or COVID-19, including clinical course, morbidity and mortality outcomes.

Comment 3. Most recently, Martineau and colleagues (2017) conducted a systematic review and meta-analysis of individual participant data from randomised controlled trials (RCTs) to assess the overall effect of vitamin D supplementation on risk of acute RTI.³⁸

There is a more recent version of this work:

Vitamin D supplementation to prevent acute respiratory infections: individual participant data meta-analysis.

Martineau AR, Jolliffe DA, Greenberg L, Aloia JF, Bergman P, Dubnov-Raz G, Esposito S, Ganmaa D, Ginde AA, Goodall EC, Grant CC, Janssens W, Jensen ME, Kerley CP, Laaksi I, Manaseki-Holland S, Mauger D, Murdoch DR, Neale R, Rees JR, Simpson S, Stelmach I, Trilok Kumar G, Urashima M, Camargo CA, Griffiths CJ, Hooper RL. *Health Technol Assess*. 2019 Jan;23(2):1-44. doi: 10.3310/hta23020.

Response: As stated in the first set of response to reviewers comments provided in Oct 2020 when this comment was provided, we reiterate "We have checked this update and it appears to be the same study in two different journals, the HTA monograph and the BMJ article. The abstracts, particularly the results section, are almost identical as they refer to the same study. It is our preference to cite the BMJ paper as it was available first."

Comment 4. "Recent rapid reviews of vitamin D for treatment or prevention in COVID-19 reported no evidence that vitamin D deficiency predisposes to COVID-19, or that vitamin D supplementation is effective in prevention or treatment of COVID-19.^{40 41} However, data sources included in the rapid review were limited.⁴²" Comment: Refs. 40 and 41 are from June and are too old now; Ref. 42 is from May.

Response: At the time of writing these citations are appropriate for the cited text. As stated in response 1, updated citations will be provided when the systematic review is updated.

Comment 5. "The search strategy was developed by the information specialists in collaboration with the research team and clinical advisors. We searched MEDLINE (OVID interface), Embase (OVID interface), Cochrane Central Register of Controlled Trials, MedRxiv and BioRxiv preprint databases on 6th-8th May 2020. We searched the global research on COVID-19 developed by the WHO,⁴⁴ CEBM Oxford,⁴⁵ and the living systematic review developed by Bern University⁴⁶ on 10 May 2020. We updated the database searches on 10th June 2020 to capture articles which may have been published since the initial search was conducted."

This is a very fast moving field of research. 10 June is too long ago; the search must be updated in January 2021. Preprint servers should not be used if they have only non-peer reviewed preprints. Scholar.google.com should be used in addition to pubmed.gov. Here are some of the new articles to include (authors have not copied the 8 citations).

Association of Vitamin D Status and Other Clinical Characteristics With COVID-19 Test Results. Meltzer DO, Best TJ, Zhang H, Vokes T, Arora V, Solway J. *JAMA Netw Open*. 2020 Sep 1;3(9):e2019722. doi: 10.1001/jamanetworkopen.2020.19722.

Low plasma 25(OH) vitamin D level is associated with increased risk of COVID-19 infection: an Israeli population-based study.

Merzon E, Tworowski D, Gorohovski A, Vinker S, Golan Cohen A, Green I, Frenkel-Morgenstern M. *FEBS J*. 2020 Sep;287(17):3693-3702. doi: 10.1111/febs.15495. Epub 2020 Aug 28.

Response: As stated in the first, and second set of response to reviewers comments provided in Oct 2020 and Jan 2021. These studies were published after the search date of our systematic review, and therefore, would not have been included in our systematic review. These studies may be captured in our updated search and systematic review as described in response 1.